# MinerGuard: A Solution to Detect Browser-Based Cryptocurrency Mining through Machine Learning

Min-Hao Wu [1], Yen-Jung Lai [2], Yan-Ling Hwang [3], Ting-Cheng Chang [1] and Fu-Hau Hsu [2,*]

1 Information Engineering College, Guangzhou Panyu Polytechnic, Guangzhou 511400, China
2 Department of Computer Science & Information Engineering, National Central University, Taoyuan City 320317, Taiwan
3 Department of Applied Foreign Languages, Chung Shan Medical University, Taichung City 40201, Taiwan
* Correspondence: hsufh@csie.ncu.edu.tw

**Abstract:** Coinhive released its browser-based cryptocurrency mining code in September 2017, and vicious web page writers, called vicious miners hereafter, began to embed mining JavaScript code into their web pages, called mining pages hereafter. As a result, browser users surfing these web pages will benefit mine cryptocurrencies unwittingly for the vicious miners using the CPU resources of their devices. The above activity, called Cryptojacking, has become one of the most common threats to web browser users. As mining pages influence the execution efficiency of regular programs and increase the electricity bills of victims, security specialists start to provide methods to block mining pages. Nowadays, using a blocklist to filter out mining scripts is the most common solution to this problem. However, when the number of new mining pages increases quickly, and vicious miners apply obfuscation and encryption to bypass detection, the detection accuracy of blacklist-based or feature-based solutions decreases significantly. This paper proposes a solution, called MinerGuard, to detect mining pages. MinerGuard was designed based on the observation that mining JavaScript code consumes a lot of CPU resources because it needs to execute plenty of computation. MinerGuard does not need to update data used for detection frequently. On the contrary, blacklist-based or feature-based solutions must update their blocklists frequently. Experimental results show that MinerGuard is more accurate than blacklist-based or feature-based solutions in mining page detection. MinerGuard's detection rate for mining pages is 96%, but MinerBlock, a blacklist-based solution, is 42.85%. Moreover, MinerGuard can detect 0-day mining pages and scripts, but the blacklist-based and feature-based solutions cannot.

**Keywords:** bitcoin; browser-based cryptocurrency mining; JavaScript miner; cryptojacking; monero; machine learning

## 1. Introduction

With the growth of the Internet, harmful code is now exhibiting a fast development trend. The Internet's diversity, speed, and scope are its primary expressions. Traditional malicious code detection techniques can no longer address the prerequisites for harmful code detection. Consider the detection of dangerous code using signatures. This technique gathers recognized harmful codes. When a new detection job is available, it consistently creates specific signatures and keeps a database of such signatures. Finding matches in the signature database allows for detection. The signature database requires a lot of work to update and maintain. Since machine learning algorithms may reveal deeper correlations between input characteristics, malicious code authors can bypass such detection methods by using simple changes like obfuscation, compression, and casing. The malicious code information would extensively abuse. As a result, machine learning-based harmful code detection often demonstrates improved accuracy and may, to a certain degree, automate the study of undiscovered hazardous code. Machine learning-based malicious code detection

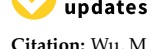



techniques have become a popular academic research area to address the problems mentioned earlier. Machine learning-based malicious code detection frequently demonstrates higher accuracy. To a certain extent, it can automate the analysis of unknown malicious code because machine learning algorithms can explore the deeper connections between input features and fully exploit the information of malicious codes.

There are several different anti-cryptojacking malware detection and defense systems available today. Browser extensions that employ blacklists to stop malicious Bitcoin mining activities are among these detection and defense measures—for instance, Nocoin, minerBlock, and other tools for detection and protection. Using blacklists by these detection and protection tools is not a suitable solution. Criminal organizations that use malware for cryptojacking alter their domain names [1,2]. It could be performed by antivirus software scanning for a set of predetermined keywords, for instance, coin, miner, and Coinhive, identifying the function name used by the web page's cryptojacked script to carry out signature matching. Additionally, it would have been claimed that this detection method is simple to bypass [3]. Researchers have suggested many high-precision detection solutions for crypto-jacking malware and existing protective measures. Examining dynamic characteristics is the foundation of most of these strategies. These include block-level analysis and dynamic detection [4–6], hardware performance counters [7], signature matching based on hardware performance counters, instruction count-based analysis, and memory events, CPU, memory, and network traffic features [8–10], runtime, network, mining-related, and browser-based features [11], and fixed thresholds. While promising and reporting excellent detection accuracy, the current detection method. There are also several difficulties to consider, such as the substantial runtime overhead of research using dynamic analytic methods. They also experience measurement errors brought on by external processes and applications' noise.

Additionally, to monitor CPU and memory events, they need administrator access. Benign Web apps also use WebAssembly, WebWorkers, and WebSockets. Therefore, current dynamic technologies may have a high risk of false positives. Additionally, using proxies, encrypted communications, and dynamically produced domain names by password hijacking writers renders detection techniques that depend on network data analysis ineffective. Finally, the end-online user's browsing experience would impact by dynamic analysis and instrument-based detection systems that operate continually in the backend.

There are several risks and descriptions of harmful programs that damage Android smartphones, according to Suleman et al. [12]. The term "cryptojacking apps" refers to one of the harmful programs on the list. It implies that Android phones and tablets would be impacted by malware and cryptojacking software. Soviani et al. [13] highlighted the value of using this method to find crypto-mining malware. They laid forth the precise procedures for developing and evaluating such systems and how to report the findings. BrenntDroid is a tool Dashevskyi et al. [14] developed for detecting mining on Android smartphones. Its method builds a dataset by gathering prospective Android mining programs and studying their behavior to ascertain if they are actual miners. Using a scikit-learn package, they filtered the dataset, and items with low variance were disregarded. Pearson correlation coefficients were used to identify and remove the characteristics with solid correlation. Using ROC and AUC, they present the results of the random forest classifier. A machine learning approach for intelligent contract security analysis could be put out by Momeni et al. [15]. They gathered information from the Etherscan dataset, including the smart contracts' source code. After feature extraction, the dataset could process using four machine learning methods: SVM, NN, RF, and DT. Next, the source code had to be built. They computed accuracy, precision, recall, and F1 to report their findings. They concluded that no one strategy works best in all situations; instead, different problems need different solutions. Another approach to anomaly identification could be put out by Huang et al. [16]. Identifying harmful nodes in blockchain networks is their primary objective. The duration between the node preparation and commit stages in various circumstances is an experimentally recorded statistic. They experimented with the

KNN, CNN, SVM, Gaussian, and Bernoulli models, marking normal and abnormal data accordingly. Agarwal et al. [17] employed temporal graph features in blockchains with fewer rights to find rogue accounts. For their assessment, they used information from the Ether scanning API. It also utilized additional sources to find and label fraudulent charges in the dataset. They determined accuracy, recall, F1 score, and MCC scores to show the experimental outcomes. Additionally, they provide cosine similarity graphs to display the similarities between malicious and benign accounts and the association between old and new harmful versions.

In September 2017, the web mining technology Eskandari [18] appeared and became popular. Later on, Coinhive mining scripts [19] were found on the web pages of the world's largest BitTorrent website, Pirate Bay. The scripts attempt to mine Bitcoin through Pirate Bay users' browsers. Many websites adopt this approach to mine cryptocurrencies without notifying their users of this behavior. Hence, the embedded scripts use users' CPU resources for mining without the users' consent. The mining operation slows down browser users' computer speed and increases the browser users' electricity bills. Abusing browser users' CPU resources in this way is called mining kidnapping [9]. This situation may become worse because even when a user leaves a mining page and closes his browser, the CPU is still used for mining without the user's knowledge. A Cisco report [10] shows that "in 2020, almost 70 percent of its customers were victims of crypto mining software".

As mining kidnapping began to spread in the wild to abuse browser users' CPU resources, many researchers started to propose methods to block web mining, such as recording the file names of mining scripts or the domain names of mining pages into a blacklist. Plug-ins such as MinerBlock, No Coin, and AntiMiner [20], or techniques that use machine learning to analyze and detect virtual currency mining scripts, such as TLSH [21], were developed. However, mining kidnapping attacks continue to exist. Additionally, new mining scripts continue emerging in the wild. Obfuscation and encryption are used to change the forms of mining scripts. The domain names of mining scripts keep changing to avoid being detected. The above approaches decrease the effectiveness of static blacklist mechanisms in mining page detection.

The CPU is a reasonably user-friendly tool for users to monitor; this system would be built on a CPU detection tool. After navigating to the "Job Manager," users may utilize the "Processors" option at the top. To see the Windows processors in the background, click "More Details" at the bottom of this Table To sort by CPU consumption, locate the CPU column towards the top of the Handler tab and click on it. Malware is likely creating the issue by impersonating a typical Windows processor if CPU processor difficulties continue using many resources. Some malware masquerades as well-known names in the task manager, such as "Cortana.exe" or "Runtime Broker," and consumes CPU and GPU bandwidth for various reasons (such as virtual currency mining). There are instances when it could not utilize mining but still uses a significant amount of CPU power, such as while playing certain games, using streaming or video-based programs, conducting antivirus scans, or browsing through several browser tabs. If you often experience excessive CPU consumption, you should shut down any background applications and open accounts. Then, check the task manager to see if anything has changed. When multitasking, a high CPU consumption is typical. Modern CPUs divide the processing power across many processor cores to handle multiplexed circumstances while using several instruction sets. If you want to achieve significant efficiency while using CPU-intensive software like Adobe Premiere, you may need to employ all of the cores that are now accessible.

This paper proposed a solution called MinerGuard to protect browsers against mining kidnapping attacks. We implemented MinerGuard as a Chrome plug-in to evaluate its effectiveness and efficiency. Because mining operations require much computation, these operations consume plenty of electric power. Based on this phenomenon, the mining page identification mechanism of MinerGuard uses the CPU resource consumption of a browser when rendering a web page as a criterion to decide whether the web page contains a mining script. MinerGuard uses machine learning techniques to distinguish

between regular web pages and possible mining pages. To demonstrate the effectiveness of MinerGuard, we conducted extensive experiments. We used 140 websites without mining scripts and 75 websites with hidden mining scripts as a dataset to train our machine learning model. Then, 800 websites were used to evaluate the effectiveness of MinerGuard. Experimental results show that MinerGuard is more accurate than the blacklist mechanism, and MinerGuard does not require manual updates, while the blacklist mechanism needs to update its blacklist constantly. Section 5 gives detailed experiment results.

In summary, MinerGuard has made the following contributions:

- Compared with the blacklist mechanism, MinerGuard does not need to update its data frequently, saving human resources and cost.
- MinerGuard has an accurate detection rate of 99%.
- MinerGurad allows a browser user to dynamically decide whether his browser can visit a web page that contains mining JavaScript code. Some websites ask browser users whether they can use the users' hosts to mine cryptocurrencies. If a user agrees to let a website perform the mining, MinerGuard will not block related web pages, but the blacklist mechanism still secures all associated web pages.

The rest of this paper is organized as follows. Section 2 describes the background knowledge required to study this paper. Section 3 discusses related work, which introduces other related defense mechanisms of Browser-Based Cryptocurrency Mining. Section 4 describes MinerGuard's system architecture and implementation details. Section 5 evaluates the detection accuracy of MinerGuard. Section 6 introduces the advantages and disadvantages of MinerGuard. This section also discusses the limitations and prospects of MinerGuard. Section 7 concludes this paper.

## 2. Background

This section describes some background knowledge, including Browser-Based Cryptocurrency Mining, BitcoinPlus.com, Coinhive, Monero, Cryptojacking, Machine Learning, Artificial Neural Network (ANN), and Browser Extensions. The previous knowledge includes the emerging services for Cryptocurrency mining and the methods used by MinerGuard for mining page detection.

### 2.1. Browser-Based Cryptocurrency Mining

In 2011, while Bitcoin was still in its infancy and was cheap and easy to mine, a service called BitcoinPlus.com [22] was published, the earliest browser-based cryptocurrency mining technology. Since that time, browser-based cryptocurrency mining has emerged.

(1) BitcoinPlus.com: As shown in Figure 1, BitcoinPlus.com uses JavaScript for centralized mining, and site owners can sign up for services and embed these scripts into their web pages. When a user browses a web page in which the mining script was embedded, the mining script will achieve the cryptocurrency mining process by using the CPU resources of the user's computer.

Although BitcoinPlus.com mined Bitcoin at the time, the service was in vain because of the low mining capacity and high power requirements. With the popularity of ASIC hardware specifically useful for mining, this service appears to be much less efficient.

Despite these benefits, browser-based cryptocurrency mining does not disappear because the goal of this approach is not to achieve mining on a single computer. The purpose of the service is to focus on many users to achieve excellent mining capabilities. This approach makes the mining system a decentralized one and has good scalability. In other words, any device that can execute JavaScript and connect to the network can participate in mining cryptocurrencies, and higher website traffic means higher revenues, which was the idea that prompted Coinhive to evolve.

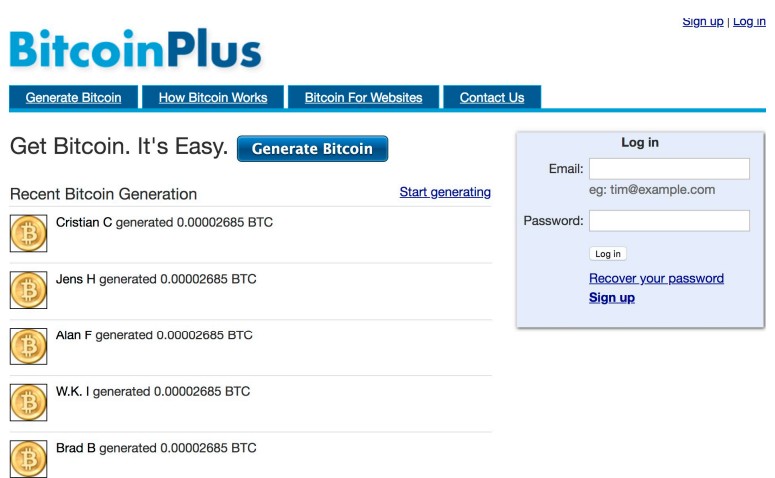

**Figure 1.** BitcoinPlus.com.

(2) Coinhive: Coinhive [18] proposed a browser mining script in September 2017 that was very similar to BitcoinPlus.com. The difference between Coinhive and BitcoinPlus.com is that Coinhive mines Monero, and BitcoinPlus.com mines Bitcoin. This technology is used to replace advertising revenue for websites. The first website that was found using Coinhive is Pirate Bay. It is a website for storing, sorting, and searching BitTorrent files. Pirate Bay is also known as the largest BitTorrent website. This site is quite suitable for browser-based mining because it takes a long time for users to search and download BitTorrent files. However, Pirate Bay did not inform its users that their computer resources were used for mining. Many users felt uncomfortable when they discovered that Pirate Bay mined in the dark and slowed down their computers.

In the past, when browsing the Internet, advertisements often affected users' perceptions about related web pages. As the Pirate Bay used mining to replace the benefits of advertising, many websites began to embed mining scripts in their web pages following the practice of Pirate Bay.

Since the advent of Coinhive, a series of competitors have emerged. For example, Crypto-Loot, Coin-Have, and Mine- MyTraffic have also launched a service called PPoi in China. Microsoft's Malware Protection Center also claimed that they discovered a new cryptocurrency mining script, one of which is CoinBlind and the other is CoinNebula.

### 2.2. Monero

Monero [23] is an open-source cryptocurrency created in April 2014 that focuses on privacy, decentralization, and extensibility. Why does Coinhive choose to mine Monero instead of mining Bitcoin? The reason is that Monero is not the same as other virtual currency algorithms, and Monero uses the Cryptonight algorithm. This algorithm is very computationally intensive and slow to calculate. It is an algorithm designed specifically to use the CPU to calculate. Computing with GPUs is only twice as efficient as CPU. This makes Monero very suitable for computing with CPU resources through the browser.

### 2.3. Cryptojacking

With the increasing popularity of browser mining scripts, many web pages have begun to use web mining instead of placing web ads. However, many web pages use the CPU resources of their visitors without the visitors' consent. This behavior causes many users to experience slower computer speeds when browsing these web pages, affecting their operations on their computers. Secretly using a browser user's computer resources for mining to earn profits, known as Cryptojacking [18], has become an attack type that consumes the user's computation resources without the user's consent. As shown in Figure 2, the CPU usage is very high when browsing web pages with embedded mining scripts.

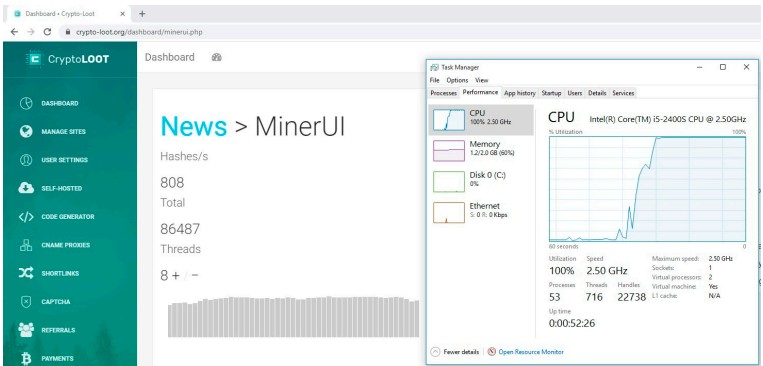

**Figure 2.** CPU usage when browsing a mining web page.

### 2.4. Machine Learning

Machine Learning [24] is the research of computer formulas that enhance automatically with experience. It is a subset of expert systems and includes numerous disciplines, including likelihood theory, stats, computational complexity concepts, etc. Machine learning algorithms develop a design based on example information, known as "training data," to make predictions or choices without being explicitly configured [25]. It would also anticipate enabling the computer to learn and distinguish the information autonomously. Machine learning extensively uses data mining, computer vision, natural language handling, biometrics, internet search engine, speech and handwriting acknowledgment, approach games, and robotics [26,27].

### 2.5. Artificial Neural Network

Artificial neural networks (ANN), referred to as neural networks (NN) or connectionist systems, are computing systems slightly influenced by the organic neural networks that make up pet minds [28]. An ANN is based upon a collection of linked systems or nodes called artificial neurons, which loosely model the nerve cells in an organic brain. Like synapses in a physical mind, each link can transfer a signal to other neurons. Artificial neural networks that receive a call then refine it and can signal neurons connected to it [29]. Moreover, an ANN can change the interior framework according to exterior details. It is a flexible system in addition to it having a knowing function.

Figure 3 shows an ANN system that includes an input layer, a hidden layer, and an output layer. The input layer is responsible for receiving external information. The hidden layer calculates and processes the input information. In processing the data, the possibility of the result is calculated through the activating function, which is called the weight distribution. The output layer is the cognitive result of the input signal for this neural network model.

### 2.6. Browser Extension

A portion of MinerGuard is implemented in a browser extension that is a small software program that enhances the user experience when using the browser. They allow users to modify the browser's features and behavior based on individual needs or preferences. They are written using standard web technologies, JavaScript, HTML, CSS, and some specialized JavaScript APIs. In addition, the extensions can add new features to the browser or change the look or content of a particular web page.

This paper's main reason for choosing Google Chrome when developing MinerGuard is that the Chrome Extension API is more mature than Firefox's WebExtensions. In WebExtensions, many corresponding APIs are still under development, such as the system API and processes API used to observe system resources in MinerGuard.

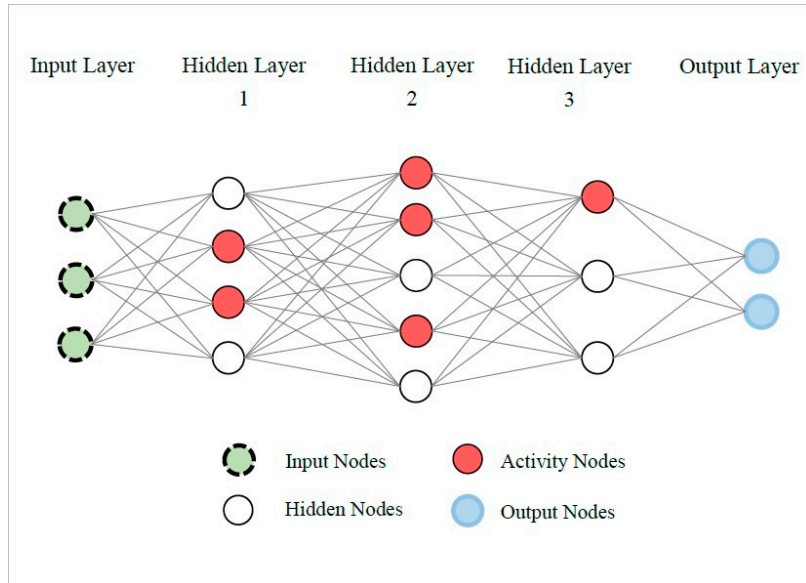

**Figure 3.** Artificial neural network system.

### 3. Related Work

This section introduces some related work about the mechanisms to block mining scripts coming from web pages. With the abuse of browser mining scripts, mining kidnapping has gradually gained security researchers' attention. Diverse methods to prevent computer resources from being abused for mining cryptocurrencies have been proposed. These methods include blacklist-based filters, feature-based detectors, and machine learning mechanisms.

The most recent research on in-browser cryptojacking detection is presented in this section. In Tekiner et al. [30], there were two cryptojacking datasets, 45 notable cryptojacking attack events, and a survey approach for detecting cryptojacking malware. They use static, dynamic, and hybrid methods to describe similar processes. Static techniques scan or crawl known malware for signatures in script code using signatures. It examines opcodes and script code of the machine-level binaries. It is also a hashing method to combat cryptojacking. Dynamic approaches look at network traffic and computational resources, such as processors/CPUs, memory, disk, power, and others. These methods may record any behavioral changes and assist in avoiding techniques like scripting and throttling. A detector based on this measures CPU consumption. There was the offer of a host-based strategy in contrast. The authors present a host performance-based strategy based on counter aspects such as CPU, memory, network utilization, and active processes on the host and network flow-based characteristics such as inbound/outbound traffic from ports 80 and 443, as used by the Stratum mining protocol. WebTestbench, another dynamic tool, uses system resources, energy use, network traffic, device temperature, and user experience, while other techniques, like those described in [31,32], examine the CPU use and patterns of JS and WASM code execution. The WASM code's execution mode and CPU use are identified. MINOS [33] uses image-based classification and deep learning techniques to distinguish benign and cryptojacking opcodes, such as those with WASM scripts. These advanced techniques are outlined in Table 1, along with their reported characteristics, datasets utilized, classifiers/methods claimed performance, and method constraints.

**Table 1.** Compared to the CPU use and methods.

| Technique | Method | Datasets | Performance/ Results |
|---|---|---|---|
| Carlin et al. [31] | RF | VirusShare OpenDNS | Acc = >99.0% |
| Bursztein et al. [32] | CNN | Alexa | Acc = 98.7% |
| Naseem et al. [33] | CNN | PublicWWW | Acc = 98.97% |
| MinerGuard | CNN | Alexa, mining web pages, | Acc = 99.0% |

### 3.1. Blacklist-Based Filter

Among the related work, we can find presently that most of the proposed solutions' prevention and control principles are based on blacklist filtering. Although this kind of method is popular, it usually faces the problem that the update speed of the blacklist cannot keep up with the speed of the generation of new mining scripts or the problem that a mining script may use an obfuscation method to avoid being detected. As a result, blacklist mechanisms cannot immediately block the latest mining scripts.

(1) MinerBlock: MinerBlock [34] is a plug-in for Google Chrome released by Ismail Belkacim, as shown in Figure 4. Currently, there are about 160,000 users. MinerBlock uses two different methods to prevent mining scripts. The first is to use a blacklist mechanism. When the browser loads a web page, if the domain name or a JavaScript file is on the blacklist, the domain name and the JavaScript file will trigger the MinerBlock blocking action, which prevents users from accessing the content. The above method is the traditional method adopted by many ad blocking and mining blocking programs. Another more effective way to avoid mining scripts is to detect the potential mining behavior of a hand and prevent it from running when the JavaScript file is loaded. This method blocks JavaScript from running through a proxy.

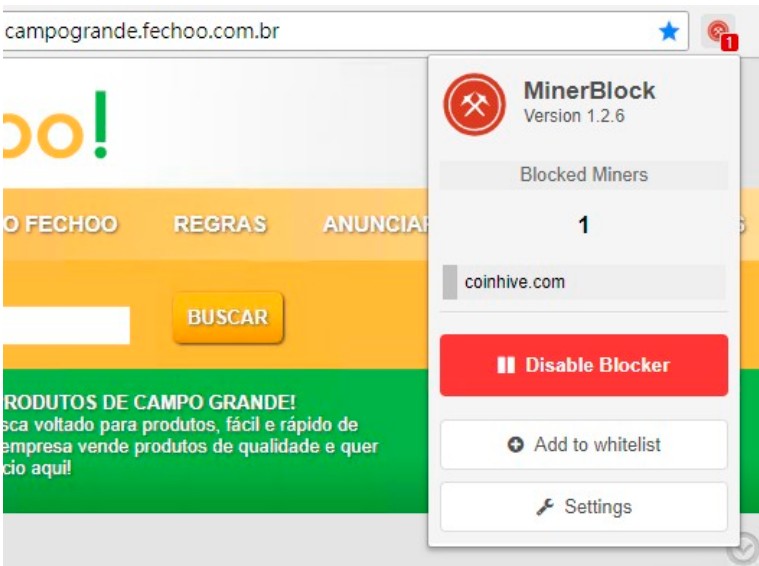

**Figure 4.** MinerBlock.

In addition, MinerBlock provides the whitelist function, which makes it possible to move the domain names or JavaScript files from the blacklist to the whitelist as exceptions. The next time the users refresh the web pages, they will not be blocked. MinerBlock can also let users maintain their blacklists by themselves, as shown in Figure 5.

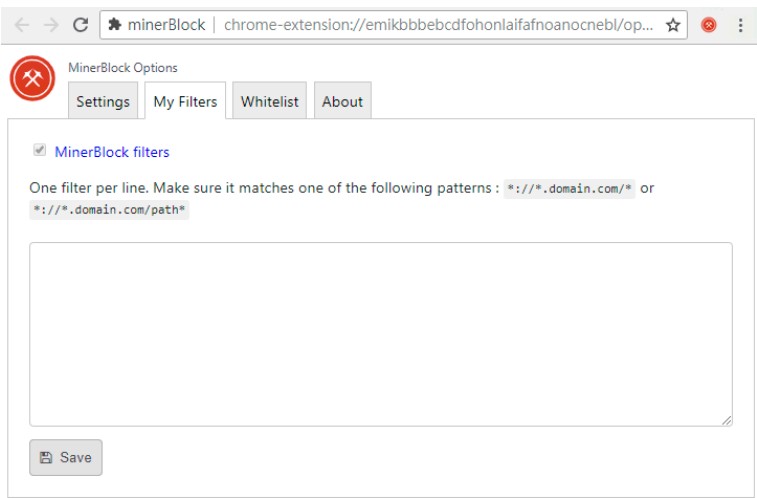

**Figure 5.** MinerBlock filter settings.

(2) No Coin—Block miners on the web!: No Coin is also a Google Chrome plug-in, an open-source project by Rafael Keramidas. It currently has about 560,000 users and is the most used mining blocker on Google Chrome. No Coin works by blocking all domain names and files in the blacklist.txt file. In addition, No Coin also provides the whitelist function and allows users to set the duration of the whitelist. However, No Coin does not provide the blacklist function.

(3) AntiMiner—No 1 Coin Minerblock: AntiMiner [20] is similar to MinerBlock and No Coin, a plug-in released by Tunghobrens that uses a blacklist and has been proven to block mining scripts such as Coinhive and JSECoin successfully. When AntiMiner detects a mining script, it will notify the user of the number and the filename of the mining script file. In addition, the AntiMiner provides the ability to add the mining script to the history list. As shown in Figure 6, AntiMiner records all blocked mining pages and presents them to the user for querying. AntiMiner also utilizes a whitelist mechanism that allows users to browse domain names and JavaScript files originally on the blacklist. Finally, AntiMiner provides the ability to maintain a customized blacklist.

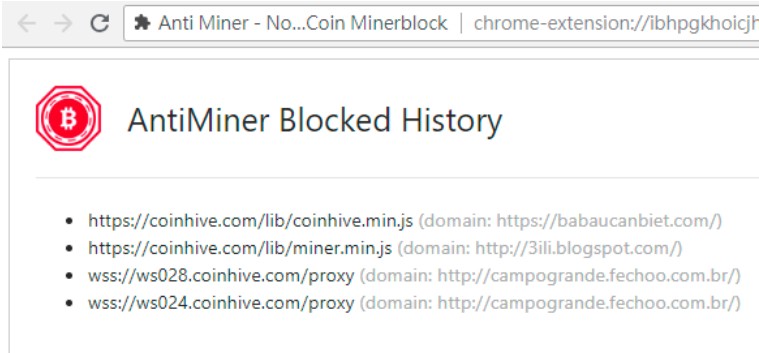

**Figure 6.** AntiMiner blocked history.

*3.2. Feature-Based Detector*

Alaeiyan et al. [35] proposed a technique by examining static trademarks, usually used for other malware, to identify cryptojacking. A similar approach has been sent by Razali et al. [36] to execute an application internet browser extension called CMBlock, which can spot and obstruct mining scripts on a website. The suggested solution combines blocklisting as well as signature-based habits detection approaches. This kind of strategy can find unidentified domain names that have not been detailed on the blocklist. Hong et al. [2] presented an additional behavior-based detector called CMTracker,

which works by determining a set of intrinsic qualities of cryptojacking scripts, such as the duplicated hash-based calculations and the regular telephone call pile.

*3.3. Machine Learning Mechanism*

Based on our study, Trend Micro published Trend Micro Locality Sensitive Hashing (TLSH) [21] in 2018, the first application of Machine Learning techniques that can quickly compare and judge similarities of mining behaviors. Given a string with a minimum length of 50 bytes, TLSH can generate a Hash for comparing similarities. Objects with a higher similarity will have higher Hash similarity to each other. Trend Micro generated the Hash from the collected virtual currency mining virus through TLSH technology and clustered similar samples. Furthermore, these clusters of samples were analyzed to identify the features of these virtual currency mining viruses, and more files with these features could be identified.

Carlin et al. [31] likewise suggested an algorithm using Machine Learning techniques for cryptojacking discovery. The writers provide a browser-based crypto-mining detection technique by assessing the vibrant opcode. The presented model can differentiate between crypto-mining sites and benign weaponized websites, such as safe areas in which the crypto-mining code has been infused, the deweaponized crypto-mining websites such as crypto-mining sites to which the begin() call has been removed, and real-world benign areas. Liu et al. [37] suggested an approach to spot the browser's harmful mining behavior. They designed a device to extract asynchronously and immediately classify the lot picture and stack features using Recurrent Neural Networks (RNNs). They conducted an experiment in which 1159 destructive examples were analyzed, and the outcomes reveal that the precision of the recognition of the original mining examples is 98% if the pieces are not encrypted and 93% if they are. Reference [38] suggested CapJack, a maker learning-based discovery mechanism able to detect in-browser dangerous cryptocurrency mining tasks. They created a machine-learning algorithm named CapsNet that simulates biological neural organization. CapJack uses system features such as the CPU, Memory, Disk, and network usage, applying a host-based option with a discovery rate of 87%. Caprolu et al. [39] suggested Crypto-Aegis that cannot just find mining habits yet likewise determines actual cryptojacking tasks with the machine learning modern technology. However, it is designed as a network-based system that spots and recognizes cryptographic operations by analyzing the accumulated network website traffic that stands out from other device learning-based devices. It carries out a detection accuracy of 96%. Nonetheless, the effectiveness of discovery might be affected by numerous cryptocurrencies and VPN items.

**4. System Design**

Section 3 shows the problems that other mining kidnapping solutions face. Hence, MinerGuard wants to achieve the following goals to eliminate these problems.

(1) MinerGuard expects to improve the accuracy of detecting mining scripts.
(2) MinerGuard expects to identify and block zero-day mining scripts instantly.
(3) MinarGurad expects to save maintenance human resources.

MinerGuard was designed based on the observation that mining JavaScript code consumes a lot of CPU resources because it needs to execute plenty of computation. Hence, the fundamental detection principle of MinerGuard is that abnormally high CPU usage of a web page represents a high possibility that the web page contains a mining script. After all, the mining code is CPU-bound code.

Figure 7 shows the execution flow of MinerGuard, which was designed based on the above principle. When a user browses a web page, MinerGuard observes the CPU usage used by the web page. It sends the CPU usage data to the machine learning model trained by MinderGuard, then uses machine learning to classify the web page to determine whether the web page contains a mining script. If the web page contains mining script code, MinerGuard prevents the mining script from running immediately.

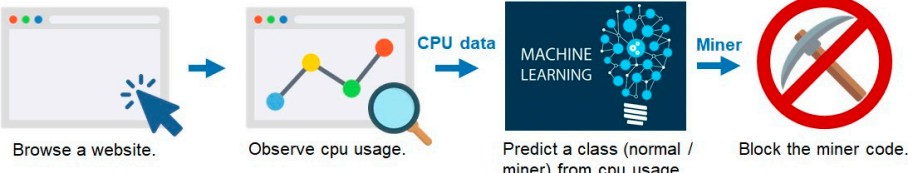

**Figure 7.** Execution flow of MinerGuard.

*4.1. System Structure*

Figure 8 shows the system structure of MinerGuard. The figure shows the major components, execution flows, and data flows of MinerGuard. MinerGuard consists of six major components, a System Information Collector, a Format Converter, an Artificial Neural Network Model, a Model Trainer, a Miner Blocking Controller, and a Database. Section 4.2 gives a detailed description of these components. These six central components are further divided into three parts, the client side, the server side, and the database.

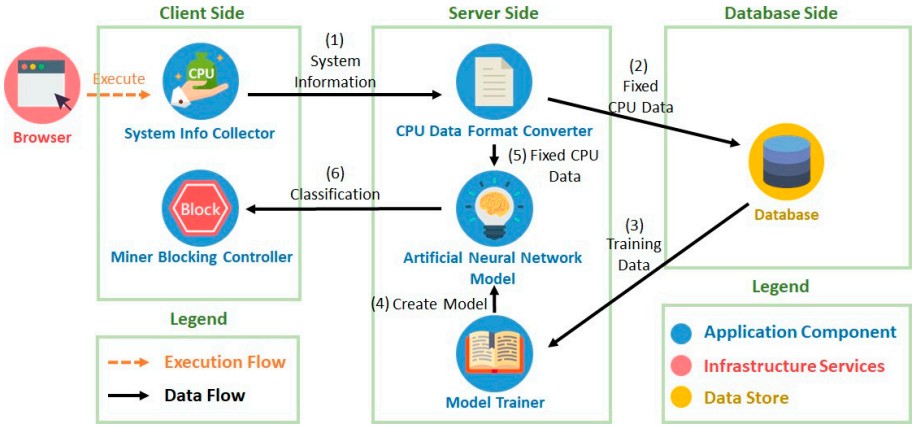

**Figure 8.** System structure.

The client side consists of a System Information Collector and a Miner Blocking Controller. The client side is implemented as a Chrome extension. The server side consists of a Format Converter, an Artificial Neural Network Model, and a Model Trainer. The server side is implemented as a program that communicates with the client and the database. The server side and the database can be on the same host where the client-side resides. The server-side and the database can also be located on a different host and serve multiple client sides on distinct hosts. Under the above structure, the server-side can collect and analyze data from various client sides, increasing the mining page detection accuracy of MinerGuard. Moreover, multiple clients using the same server side can share their previous web page analysis results, thus increasing the performance of MinerGuard.

When a user opens the browser, MinerGuard automatically executes in the background until the browser is closed. The data flows in MinerGuard can be divided into six flows, each of which is described as follows:

(1) MinerGuard observes the browser's system resource usage in the background. Whenever the amount of system data sufficient for identification by the artificial neural network model is collected, the system data are passed to the Format Converter for conversion.

(2) The Format Converter extracts the CPU usage data from the original system resource data, converts it into CSV format, and stores it in the database.

(3) Transfer the CPU data collected in the database as training data to the Model Trainer.

(4) Set the Artificial Neural Network Model parameters according to the training data to train the best model. Generate an Artificial Neural Network Model responsible for identification.

(5) The converted data are sent to the Artificial Neural Network Model, classified by the model, and the converted CPU data are classified as value 0 or 1; 0 means the web page has a malicious mining script.

(6) The value of this classification is transmitted back to the Miner Block Controller on the client side. If it is classified as 0, the operation of the mining script of this web page is blocked. If the classification is standard, no action is taken.

In short, MinerGuard starts working when a user uses a browser to visit a web page. The System Information Collector then continuously observes the CPU usage in the background and sends the collected CPU usage data to the Format Converter of the server-side. The Format Converter converts the original CPU data into the input format of the Artificial Neural Network Model, a pre-processing action. The data are stored in the database. Then, the converted data are sent to the artificial neural network model for prediction, and the predicted result is transmitted back to the user. Based on the development, the component Miner Block Controller decides whether the browser should stop the execution of script code on the web page or not.

*4.2. Main Components*

This subsection introduces the significant components of Miner-Guard, which consist of six main parts: (1) System Information Collector, (2) Format Converter, (3) Artificial Neural Network Model, (4) Model Trainer, (5) Miner Blocking Controller, and (6) Database.

(1) System Information Collector: System Information Collector obtains the number of processors in a system via chrome.system.cpu.getInfo. The resource changes of each processor in the system are then monitored by chrome.processes.onUpdated.addListener. This method updates resource changes over time.

Chrome.processes.onUpdated.addListener returns a dictionary object that is indexed by the process ID and contains the system resources of the running process. Therefore, we can obtain the paging ID, title, CPU usage, and memory usage from this dictionary. The amount of CPU usage is the percentage used in a single CPU. Memory is the amount of memory used per second, in bytes, where we convert it to megabytes. Finally, this component adds the CPU usage of all processes and divides by the number of CPU cores to observe the total CPU usage. Figure 9 displays the system details gleaned from the Chrome extension's APIs.

```
15      //Get the number of processes
16      chrome.system.cpu.getInfo(function(cpuInfo) {
17          numOfProcessors = cpuInfo.numOfProcessors;
18      })
19      //Update resource of system
20      chrome.processes.onUpdated.addListener(receiveProcessInfo);
```

**Figure 9.** Obtain system information through Chrome extensions APIs.

The System Information Collector observes and records the various resource usage of each process for three minutes. Then, the System Information Collector converts the observed data into the CSV format and sends them to the Format Converter using the HTTP POST method. Figure 10 will display the memory and CPU consumption.

(2) Format Converter: After the Format Converter receives the CSV data transmitted by the System Information Collector, it converts the data to the ndarray format in NumPy. NumPy is an extensive library of Python. NumPy supports many-dimensional arrays and matrix operations, which are suitable for mathematical operations or operations that need to handle a large amount of data. Converting to ndarray in this component is conducive to the prediction performed in the next phase.

(3) Artificial Neural Network Model: We made use of Keras running on top of TensorFlow in our execution to establish our Artificial Semantic network Version. Keras [40] is an

open-source neural network collection written in Python. It is designed to enable quick experimentation with deep neural networks and focuses on being straightforward, modular, and extensible. It supplies a higher-level, more user-friendly collection of abstractions that make it simple to create deep understanding designs despite the computational backend. On the other hand, TensorFlow [41] is a complimentary and open-source symbolic math collection and is likewise used for artificial intelligence applications such as semantic networks. Its versatile design permits the simple deployment of calculation throughout different platforms (CPUs, GPUs, TPUs) as well as from desktop computers to collections of servers to mobile and edge gadgets.

```
33  function receiveProcessInfo(processes) {
34      for (pid in processes) {
35          var tab_process = new Object();
36          tab_process['tab'] = processes[pid].tasks[0].title;
37          tab_process['tabId'] = processes[pid].tasks[0].tabId;
38          tab_process['cpu'] = processes[pid].cpu;
39          tab_process['memory'] = (processes[pid].privateMemory / 1024 / 1024);
40      }
41      totalCPU /= numOfProcessors;
42
```

**Figure 10.** Obtain CPU usage and memory usage.

To load the pre-trained model and make predictions, we took 150 pieces of data from the variety and used them as an input to the Artificial Neural Network Design. The artificial Semantic network Model executes weight estimations based upon the features of the input information to obtain the chance that a website is a regular one or a mining web page. Lastly, the forecasted possibility worth is stabilized to 0 or 1; 0 indicates that a website is a typical web page, as well as 1, which implies that a mining script is embedded in the website. Figure 11 will demonstrate how to tell whether the website has mining code placed in it.

```
19      # split into input variables (x_test)
20      x_test = dataset[:,0:150]
21
22      # Load model from h5 file
23      model = tf.contrib.keras.models.load_model('model/miner_model.h5')
24
25      # predict
26      prediction = model.predict_classes(x_test)
27
28      # Result format = [[0]], 1 = miner, 0 = normal.
29      print(prediction[0][0])
30
```

**Figure 11.** Identify whether the website is embedded mining code or not.

(4)  Miner Blocking Controller: Based on the result predicted and sent by the Artificial Neural Network Model, the Miner Blocking Controller performs corresponding operations. Result 0 means that the related web page is regular; hence, the Miner Blocking Controller lets the browser process the web page as usual. Result 1 means that the web page is a mining page. In this case, the Miner Blocking Controller pops up a window to inform the browser user that the web page contains a mining script and asks whether the user wants the browser to continue processing the web page. In other words, the Miner Blocking Controller lets the user decide whether to stop browsing the mining page or continue processing the mining page.

(5)  Model Trainer: The Model Trainer trains the Artificial Neural Network Model of MinerGuard using a large amount of web page resource usage data collected by the System Information Collector. Our model uses an artificial neural network model. The input training data consist of pairs of datasets and tags. A dataset consists of the successive CPU usage data of a web page. A label value is either 0 or 1. A result

of 0 means a related web page is a normal one, and 1 means a related web page is a mining page.

Our Artificial Neural Network Model has five layers: the input layer, the output layer, and the three hidden layers. The task of these three hidden layers is to learn the features of the CPU data automatically. After training with large amounts of data, the model can be exported. Therefore, it is not required to start preparing the model from the beginning for later predictions.

The operation steps of the Model Trainer are described as follows. As shown in Figure 12, the neural network model randomly activates several nodes in each hidden layer. An activated node means that the transmitted message will be considered a more important message and a major factor affecting the judgment result.

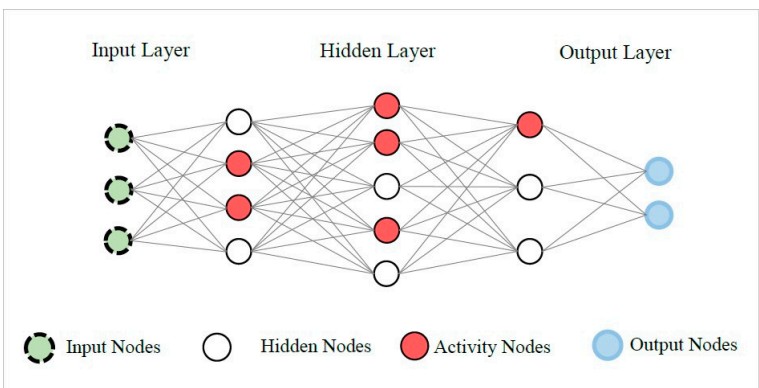

**Figure 12.** Activating some hidden nodes.

When the neural network receives input data, including a dataset and a label, the dataset will be sent to the hidden layer from the input layer. The trainer will calculate its output based on the different weights of the activated nodes and the deactivated hidden node, and the result node will be compared with the label of the input data finally. As shown in Figure 13, if the prediction result does not match the label, the node that is activated by this prediction needs to be readjusted.

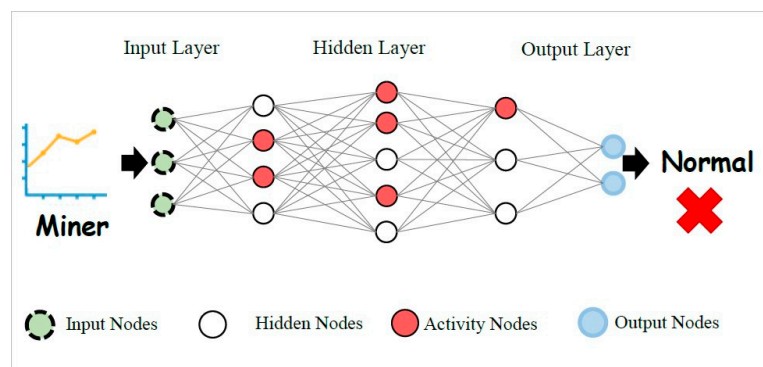

**Figure 13.** ANN model failure prediction.

As shown in Figure 14, we can predict the correct result by constantly adjusting the activated nodes and gradually identifying the important features in the data. We used Keras to create and educate our Artificial Neural Network Model in our application. First, we started a direct stacking design, an ANN design without branches. This indicates that each layer has just one piece of input data and one piece of result data, and each layer is the input information of the following layer. In the following action, we add four surprise layers and set the variety of nodes suggested in each layer, the input information for the triggering feature, and the dimension of the input information indicated in the first layer.

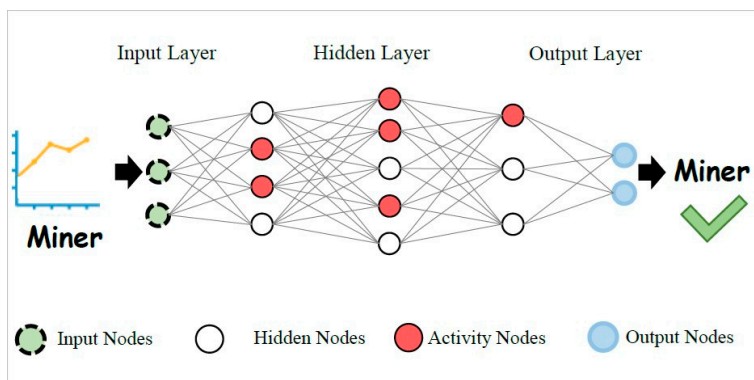

**Figure 14.** ANN model successful prediction.

Figure 15 depicts the updated artificial neural network model for MinerGuard. Sequential is a linear stacking model, meaning it is branchless. Only one input and one output are present in each layer, and each layer's work serves as the following layer's input. Then, four further hidden layers are included, defining the amount of the input data to be supplied in the first layer, the incentive function to utilize, and the number of nodes implied in each layer.

```python
# create model
model = Sequential()
model.add(Dense(80, input_dim=150, activation='relu'))
model.add(Dense(20, activation='relu'))
model.add(Dense(12, activation='relu'))
model.add(Dense(1, activation='sigmoid'))
```

**Figure 15.** Create ANN model.

Before starting the model training, the learning configurations used by the compiler must be adjusted. The compiler has three parameters: the loss (objective) function, the optimizer, and the metric. Let us give an example. As shown in Figure 16, the loss function is set to binary cross-entropy, a function for solving the binary classification problem. We choose adam as the optimizer, which is the best choice in the case of fewer data. Regarding the metric, we set accuracy to handle the classification problem in MinerGuard.

```python
# Compile model
model.compile(loss='binary_crossentropy',
              optimizer='adam',
              metrics=['accuracy'])

# Fit the model
model.fit(dataset, tag, epochs=150, batch_size=20)

# evaluate the model
scores = model.evaluate(dataset, tag)
```

**Figure 16.** Training ANN model.

In addition, the trainer uses the fit method for training our ANN model, which contains four parameters: dataset, tag, epochs, and batch size. The dataset and tag are our training set and target, respectively. On the other hand, the generations and batch size represent that MinerGuard performs the training with 20 samples as a batch and 150 times for each.

## 5. Evaluation

In this section, various experimental results are presented to show the efficiency and accuracy of MinerGuard. This section contains two subsections. The first subsection shows the environment setup used in our experiments. This subsection also introduces the test cases that we chose. The second subsection explains the experiments for efficiency and accuracy evaluation.

### 5.1. Experimental Environment and Test Cases

We implemented MinerGuard on Google Chrome and Windows 10. Table 2 shows our experimental environment. In our experiments, we collected 10,087 websites from Alexa's 4000 websites and 6087 mining web pages found by security researchers to train our neural network model and test our detection accuracy. For each of the 4000 of Alexa's websites, we used the home page of the website to represent the website. Since many websites register the same domain names in different countries simultaneously, as shown in Table 3, we deem domain names that are only different at the top-level country domain as the same domain name.

**Table 2.** Experimental environment.

| | |
|---|---|
| Server OS | Ubuntu 16.04 x64 |
| Client OS | Windows 10 |
| Client CPU | Intel Core i5-4200H |
| Brower | Google Chrome 69.0.3464.0 dev |
| Programming Language | Python 3.5.2 and PHP 7.0.30 |
| Machine Learning Framework | TensorFlow 1.7.0 and Keras 2.1.5 |

**Table 3.** Example of a web page with different domain names in different countries.

| URL | Country |
|---|---|
| bitcoinearningblogs.blogspot.de | Germany |
| bitcoinearningblogs.blogspot.fr | France |
| bitcoinearningblogs.blogspot.gr | Greece |
| bitcoinearningblogs.blogspot.hu | Hungary |
| bitcoinearningblogs.blogspot.lt | Italy |
| bitcoinearningblogs.blogspot.nl | Netherlands |
| bitcoinearningblogs.blogspot.pt | Portugal |
| bitcoinearningblogs.blogspot.sg | Singapore |

A website may be compromised and embedded with mining scripts. Still, later on, when we did our experiments, if the mining pages were discovered and removed from the website, the related web pages should be deemed as standard web pages, even though they were listed as mining pages by security researchers. We chose the home pages of Alexa's top 500 websites and 350 mining pages from the above 10,087 web pages to do our experiments. The remaining 9237 web pages were used to train our neural network model. Table 4 shows the samples we tested. A web page that informs its users of its mining behavior is deemed a regular web page. After all, their users know that the web page will perform mining, and the web page handles the mining code according to their users' decisions.

**Table 4.** Types of tested web pages.

| Web Page Type | Number |
|---|---|
| Mining Web Page | 150 |
| Mining Web Page with Users' Approval | 200 |
| Normal Web Page (Alexa Top 500) | 500 |
| Total | 850 |

*5.2. Overhead and Detection Accuracy*

MinerBlock, No Coin, and AntiMiner mentioned in Section 3 are protected by blacklists. However, MinerBlock uses two protection mechanisms, which makes MinerBlock more accurate than No Coin and AntiMiner. Thus, in this subsection, we use MinerBlock to represent the blacklist mechanism. Moreover, we measured the resource usage overheads introduced by a browser installed with MinerBlock and a browser installed with MinerGuard.

(1) Overhead for Handling Normal Web Pages: Figure 17 shows the CPU usage overheads of MinerGuard and MinerBlock when examining normal web pages. The solid lines in Figure 13 represent CPU usage overheads, where blue represents the CPU usage overheads of MinerBlock and yellow represents the CPU usage overheads of MinerGuard. The figure shows that the CPU usage overheads of MinerBlock approach zero most of the time. The overheads introduced by MinerBlock mainly occur when a browser completes downloading a web page because MinerBlock only checks a web page right after the page is downloaded by a browser. As a result, most of the time, the CPU usage overheads of MinerBlock are very low, and the overheads increase only when a browser downloads a web page. However, MinerGuard constantly executes in the background to collect the system resource usage of a browser; hence, MinerGuard's CPU usage overheads are higher than MinerBlock. Nonetheless, the overheads are maintained at 2%, which is trivial; thus, MinerGuard does not affect the normal operations of browsers installed.

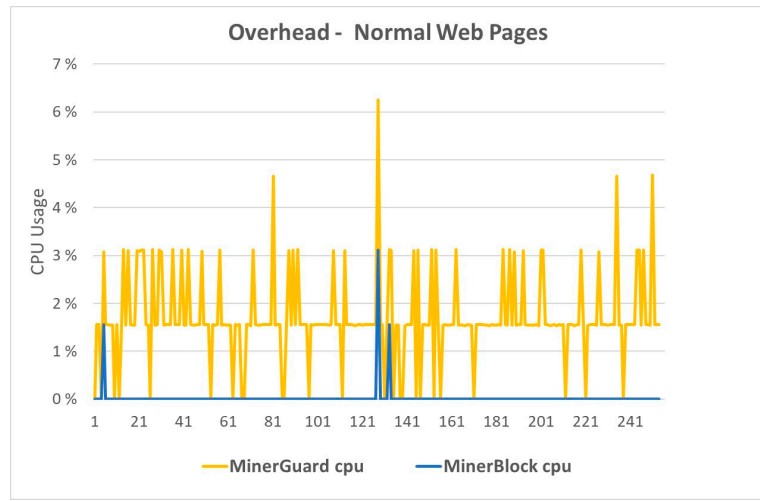

**Figure 17.** Resource usage overheads for handling normal web pages.

(2) Overhead for Handling Mining Web Pages: Figure 14 shows the CPU usage overheads of MinerGuard and MinerBlock for detecting mining web pages. The solid lines in Figure 18 represent CPU usage overheads, where dark blue represents the CPU usage overheads of MinerBlock and light blue represents the CPU usage overheads of MinerGuard. Figure 18 shows that the CPU usage overheads of MinerBlock are higher than that of MinerGuard. When handling a web page containing a script, MinerBlock needs to compare and check its blacklist. MinerBlock also needs to intercept web

requests. The above operations increase the CPU usage overheads of MinerBlock. Moreover, MinerBlock provides functions to edit its blacklist and whitelist. Both functions also increase the CPU usage overheads of MinerBlock. However, no matter what web pages MinerGuard handles, it just executes in the background to collect system resource usage. This operation only introduces about a 2% CPU usage overhead. However, classifying a web page using a neural network model and blocking the execution of a mining page add extra CPU usage overheads to MinerGuard. This is why the CPU usage overheads of MinerGuard suddenly increase a lot when it detects a mining page. Memory utilization in MinerGuard and MinerBlock is compared. The graph's dashed line represents memory use, and when it comes to finding mining locations, there is no discernible difference between MinerGuard's (yellow dashed line) and MinerBlock's (blue dashed line) memory usage.

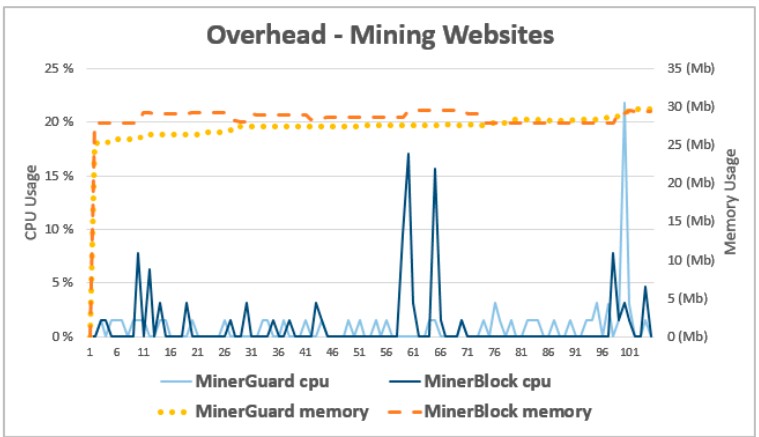

**Figure 18.** Resource usage overhead for handling mining web pages.

(3) Detection Accuracy: As shown in Figure 19, the accuracy of MinerGuard in detecting mining pages is 96%, while the accuracy of MinerBlock is 42.85%. Some web pages will inform their users of their using mining pages. These web pages execute mining scripts only after they obtain users' consent. Moreover, according to the definition of Cryptojacking, a web page is deemed malicious. Suppose the web page does not tell its users about its mining behavior. As a result, we do not classify the above type of web pages as mining pages. However, approaches that adopt blacklists block web pages in a blacklist, even if a web page informs its users of its mining behavior and its users agree to let the web page mine using their devices. In this situation, the detection accuracy of both MinerGuard and MinerBlock for normal web pages is high.

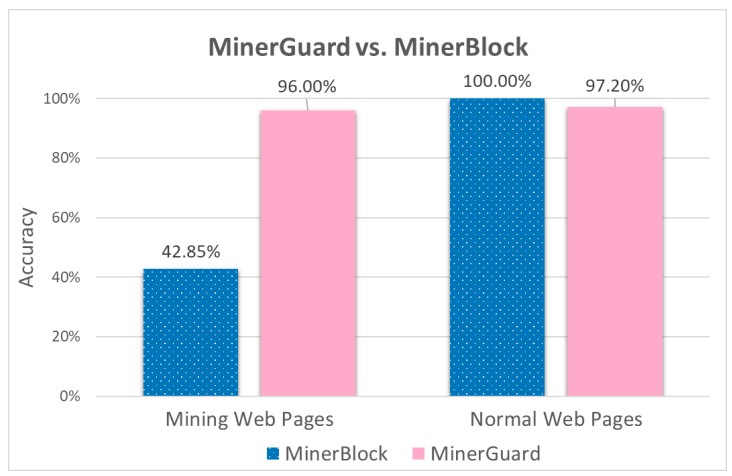

**Figure 19.** Detection accuracy for mining web pages and normal web pages.

(4)   Comparisons among Related Work: We made various comparisons among Miner-Guard, three blacklist-based methods, MinerBlock, No Coin, and AntiMiner, two feature-based methods, CMBlock and CMTracker, and two machine learning mechanisms, CapJack and Crypto-Aegis, to show the advantages and disadvantages of these techniques. The comparisons are shown in Table 5. As the mining scripts are constantly updated, the mining methods are continuously improved, and the blacklist-based and feature-based mechanisms must regularly maintain and update their blacklists. MinerGuard adopts behavior-based machine learning techniques to identify mining activities and, therefore, does not require frequent maintenance. As new mining scripts and pages are constantly developed, MinerGuard can detect these 0-day mining pages. However, blacklist-based and feature-based methods cannot see them in time because related information of 0-day mining scripts or pages needs to be added to blacklists, which costs some effort. In addition, MinerGuard will notify a user if a web page contains mining behaviors and let them decide to allow the execution of mining scripts. However, the methods mentioned earlier, other than MinerGuard, will block a mining script no matter whether a user agrees with the mining activity. When using these methods, if a user wants to agree to execute a mining script on a web page, they need additional efforts, such as manually appending hash values or rules.

**Table 5.** Comparison table.

|  | MinerGuard | MinerBlock | No Coin | AntiMiner |
|---|:---:|:---:|:---:|:---:|
| **Maintenance** | ✗ | ✓ | ✓ | ✓ |
| **0-day mining detection** | ✓ | ✗ | ✗ | ✗ |
| **Block the legal websites** | ✗ | ✓ | ✓ | ✓ |
| **Block the websites without users' approval** | ✗ | ✓ | ✓ | ✓ |

Table 5 contrasts MinerGuard's benefits and drawbacks with those of the three blacklist defense mechanisms: MinerBlock, No Coin, and AntiMiner. The blacklist protection system must periodically maintain and update the blacklist material since mining scripts are updated and techniques are improved. On the other hand, our MinerGuard does not need routine maintenance since it recognizes mining activity based on the features of the mining behavior. Our MinerGuard can identify the new mining scripts constantly being released as long as they fit the traits of regular mining activity. MinerBlock, No Coin, and AntiMiner, on the other hand, need to be manually added to the blacklist regularly and are unable to safeguard consumers promptly. Additionally, if we ask for the user's permission to cease mining operations, we will not do so immediately; instead, MinerGuard will always solicit the user's input before deciding whether to block mining scripts. Users must manually add mining scripts to the whitelist if they wish to provide their consent to mine since MinerBlock, No Coin, and AntiMiner all block mining scripts by default.

## 6. Discussion

MinerGuard currently only works on the development version of Google Chrome, as the chrome.processes API used in this study is presently being tested and is not officially open and stable. This API is required to collect CPU usage. To protect users accurately and promptly, we recommend that users install a development version of Google Chrome. Moreover, suppose a malicious mining script uses very low CPU resources to mine. In that case, our MinerGuard cannot accurately identify this type of mining script because the pattern of its CPU resource usage changes is very similar to that of a normal web page. However, shallow CPU usage is not attractive to vicious miners and does not affect users' usage of their hosts.

Furthermore, MinerGuard detects mining pages based on monitoring the CPU usage introduced by handling a web page. As a result, a mining script can still execute in a browser with MinerGuard. However, it takes a short time for MinerGuard to find a web page's abnormal CPU usage behavior. Hence, even if a mining script is executed, it can only execute for a short time.

## 7. Conclusions

Mining kidnapping has become one of the latest attack trends in the security field. Many security researchers have proposed diverse solutions to block mining pages. Blacklists are a common mechanism used in blocking mining pages. A blacklist may include the file names of mining scripts and the domain names of mining pages. However, as malicious mining spreads, new mining scripts are constantly introduced. Moreover, vicious miners also apply obfuscation or encryption to mining scripts to bypass detection. Vicious miners may also change the domain names of mining scripts. As a result, to maintain the detection accuracy of the blacklist mechanism, developers need to update their blacklists' content regularly. In this study, we present the issues that the blacklist protection mechanism faces. Moreover, some web pages ask browser users whether the users allow their mining code to be executed. These web pages execute the mining scripts only after browser users' permission, however, without manually changing the content of a blacklist or a whitelist. Even if a user wants to visit a web page that is on a blacklist, the user still cannot see the web page. MinerGuard solves the above problems. The detection accuracy of MinerGuard for mining web pages is 96%, and the detection accuracy of MinerGuard for normal web pages is 97.2%. With MinerGuard, we use machine learning to gather actual data, train on it, and then effectively build an artificial neural network model with 99% accuracy. We then use the trained model to detect mining scripts. Compared to other related solutions, MinerGuard is accurate and can detect 0-day mining scripts and 0-day mining pages. Hence, MinerGuard provides users with safe and friendly protection.

**Author Contributions:** Conceptualization, Y.-J.L. and F.-H.H.; Formal analysis, Y.-J.L.; Methodology, and F.-H.H.; Supervision, F.-H.H.; Visualization, Y.-L.H. and T.-C.C.; Writing—review & editing, M.-H.W. All authors have read and agreed to the published version of the manuscript.

**Funding:** This research received no external funding.

**Acknowledgments:** This work was completed under the auspices of the Industrial Technology Research Institute. The financial support provided by the Bureau of Energy, Ministry of Economic Affairs, Taiwan, R.O.C. under Grant No. 110-E0203 is greatly acknowledged. This work was also supported by the Ministry of Science and Technology, Taiwan, R.O.C. under Grant No. MOST 105-2221-E-008-074-MY3. This study was funded by Research on EEG signal emotion recognition and regulation strategy based on deep learning (1007/210224252) and the Innovation team of sign-oriented information perception and artificial intelligence application (1007/210224085).

**Conflicts of Interest:** The authors declare no conflict of interest.

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
