# Peer review of "MinerGuard: A Solution to Detect Browser-Based Cryptocurrency Mining through Machine Learning"

_applsci, doi:10.3390/app12199838_

Round 1

Reviewer 1 Report

paper has a lot of weaknesses that should be considered to improve this research.

- The work has old references, they would have to be updated since several works have come out in recent years.

- It is necessary to explain in detail why the techniques mentioned were used, it is said that neural networks are used but it is not mentioned why the one used was chosen.

- It would also be necessary to justify, why to make a system that is based only on the processing of CPU? and which problems it would imply in cases that are not for mining but also consume a lot of CPU processing.

- There is no mention of GPU processing at all, and it is important as GPUs have now become an alternative for heavy data processing.

- The results are only compared with a single variant (MinerBlock), but it would be necessary to be able to compare it with other like open source and free alternatives

- The difference between MinerBlock and the proposed system is very large, so a better explanation of these results would be needed, in order to understand why the difference is almost double.

- In general, the work needs to be redirected and compared with current works 2021-2022 and also with other systems, in order to be considered for publication. And also explain and justify why the authors chose the methods that are used.

Author Response

Dear reviewer:

We are very grateful for the reviewer's comments about the manuscript. According to the reviewer's comments advice, we amended the relevant part of the manuscript. Some of the reviewer's comments and questions are answered below.

Thanks, the reviewer, very much for the comments.

Reviewer 3 Report

This research methods (especially machine learning techniques) and final results are very useful, popular and very interesting.

This paper proposed a solution called MinerGuard to protect browsers against mining kidnapping attacks. Authors implemented MinerGuard as a Chrome plug-in to evaluate its effectiveness and efficiency.

Author Response

(The authors gave the same response as above.)

Reviewer 4 Report

This paper proposes a solution 26 called MinerGuard, to detect mining pages. Based Cryptocurrency Mining through Machine Learning

The cover covers a significant subject that falls within the series of the journal, but The paper has many weaknesses, in my opinion - in the following - I explain the highest shortcomings in terms of structure, objective, declaration of the problem.

From page 2 to 7 A long introduction to the topic, and there is no need to mention all these details and must be brief In the related work more discussion and analysis is required (advantages and disadvantages of the previous efforts must declared , the accuracy also of the previous effort must be listed , recent work in the area must be added The architecture of the proposed . ANN model failure prediction must be adjusted TYPES OF TESTED WEB PAGES (150-200 500) why – author must adjust these numbers The no of Mining Web Page (150) not enough

In the introduction authors claimed " MinerGuard has an accurate detection rate of 99%" that is not clear in the results
